# Aligning Language Models to User Opinions

**EunJeong Hwang**[1,2], **Bodhisattwa Prasad Majumder**[*3], and **Niket Tandon**[*3]

[1]University of British Columbia
[2]Vector Institute for AI
[3]Allen Institute of AI
ejhwang@cs.ubc.ca, bodhisattwam@allenai.org, nikett@allenai.org
*contributes equally

## Abstract

An important aspect of developing LLMs that interact with humans is to align models' behavior to their users. It is possible to prompt an LLM into behaving as a certain persona, especially a user group or ideological persona the model captured during its pertaining stage. But, how to best align an LLM with a specific user and not a demographic or ideological group remains an open question. Mining public opinion surveys (by PEW research), we find that the opinions of a user and their demographics and ideologies are not mutual predictors. We use this insight to align LLMs by modeling relevant past user opinions in addition to user demographics and ideology, achieving up to 7 points accuracy gains in predicting public opinions from survey questions across a broad set of topics[1]. Our work opens up the research avenues to bring user opinions as an important ingredient in aligning language models.

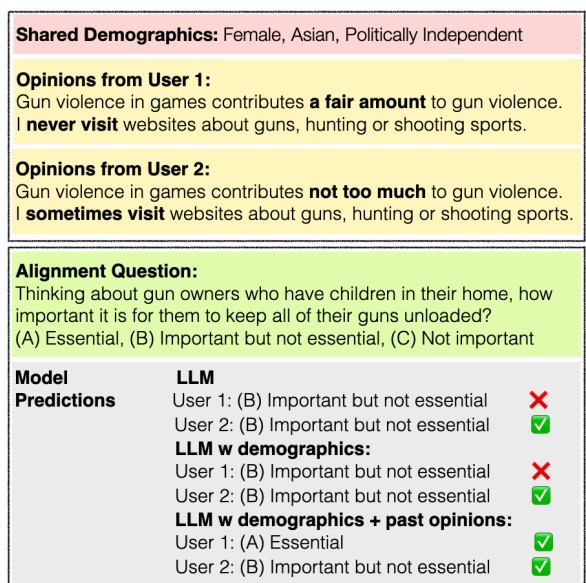

Figure 1: Opinions can vary even when two users have the same demographic traits.

## 1 Introduction

Personality is a defining feature of human beings, shaped by a complex interplay of demographic characteristics, moral principles, and social experiences (Weil, 1957; McLellan, 1989). In turn, a person's personality has a significant influence on their ability to make decisions (Lauriola and Levin, 2001; Busic-Sontic et al., 2017). Owing to the wide-scale adaptation of the large language models (LLMs) for assisting individuals in their decision-making process (Jiang et al., 2021; Gao et al., 2023a), it becomes increasingly critical to ensure that these models are aligned with the unique personalities of their users.

With lower barriers to entry, several recent works focused on prompting LLMs with persona or role-based prompts such as `Pretend you are a Democrat` (Deshpande et al., 2023; Santurkar et al., 2023). However, the extent to which these approaches align language models with users remains unclear due to the subjective nature of defining user personas. Users have nuanced opinions that can change over time and vary depending on context. While alignment with normalized user groups like religion or political inclination may be easier, LLMs continue to struggle to align with individual users or the long tail of user groups. Additionally, LLMs tend to form opinions based on their pre-training data and feedback collected from crowd workers and model designers. As a result, they exhibit low steerability, even with user groups that have major representation (Santurkar et al., 2023).

Aligning LLMs to individual and long-tail opinions has received less attention, while mostly focusing on aligning to user groups. In our analysis over PEW surveys, we found that people can share all of their demographic traits but still exhibit a large variance in their opinions, rendering the current group-based LLM alignment insufficient. This

---

[1]Project page:
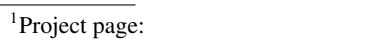
https://github.com/eujhwang/personalized-llms

paper investigates the relationship between demographic traits and individual opinions in LLM alignment. Specifically, we seek to answer the following research question:

> What do we need to align an LLM to a user: demographic traits, fine-grained opinions, or both?

The majority of the past work in NLP literature focused on aligning LLMs with normalized user groups (Santurkar et al., 2023; Majumder et al., 2019; Salemi et al., 2023). In social science studies, however, it has been shown that all users are unique even if they belong to the same broader user group, and normalizing user groups is not a true representative of a user's opinion (Chu et al., 2023; Kim and Lee, 2023). We apply insights from these social science studies to an empirical setting where we try to model individuals' opinions based on their various persona information such as demographic traits, ideological inclinations, and past opinions.

In this paper, we give a thorough analysis of public survey responses in the OpinionQA dataset (Santurkar et al., 2023) with respect to their demographics, ideology, and implicit opinions and present comprehensive experimental results using the GPT-3 model with various combinations of inputs (i.e., demographic, ideology, user's past opinions). Through our dataset analysis, we found that users' opinions and demographics do not necessarily correlate with each other. Our experimental results show incorporating both user opinions, demographics, and ideology, results in significant gains of up to 7 points in QA accuracy for certain topics, and utilizing the most relevant past opinions helps the model to pinpoint the more accurate answers for the users.

## 2 What makes a persona?

We present a study on various components that makes a personality (in short, persona) of a user. We use the OpinionQA dataset, which contains 15 topics, and each topic contains an average of 100 questions and 5K users (Santurkar et al., 2023).

### 2.1 Demographics

The dataset records eight demographic information of a user: region, sex, age, education, race, citizen, marital status, and income. These are the markers of social experience that a user is most likely to go through. For example, the social experience

can be determined by the region a user belongs, or their age determines whom they socialize with on a regular basis. However, this runs with the risk of stereotyping (i.e., an old individual is less likely to mix with younger people or they are conservative in thinking). We later show that demographic information is not enough to model an individual.

### 2.2 Ideology

Ideology is formed by an individual understanding of politics and economics. In our dataset, we have each subject's political affiliation and inclinations toward well-known political ideologies (e.g., conservative, liberal). We use this information as an individual's ideology.

### 2.3 Opinions

OpinionQA uses a well-established method of capturing human opinions from public opinion surveys. In these surveys, subjects are asked to answer subjective questions that reflect their unique opinions and what makes them different from other individuals. Figure 1 shows an example of opinions that a user provided during a survey.

### 2.4 Deriving insights from public surveys

We derive insights from the OpinionQA dataset, where we analyze the degree of agreement in user's opinions where they same demographics and how this agreement varies across topics. This statistical analysis generates useful insights that we later use for our modeling approach. We also look for similar (dis)agreements in opinions when users have the same ideologies.

**Opinions differ despite same demographics** We first take all pairs of users sharing the same demographics and compare their opinions. To calculate the agreement score between users, we utilize Cohen's kappa coefficient (Cohen, 1960), which ranges from $-1$ to 1. Even though two users share the same demographics, agreement scores on the implicit opinions are gathered around 0.5 (Figure 2). This shows that solely relying on demographic information is not enough to personalize the model, and users' implicit opinions can play a critical role in personalization.

**Opinions differ across topics** In Figure 2, we also show the topic-wise agreement scores. On certain topics, including Family & Relationships and Guns, users exhibit relatively higher agreement

scores. On the other hand, for some topics, including Race and America in 2050, users have lower agreement scores, indicating that certain topics may have larger variability in terms of user opinions. We later analyze if this variability appears in a model's predictive performance when it is used to predict user opinions across different topics.

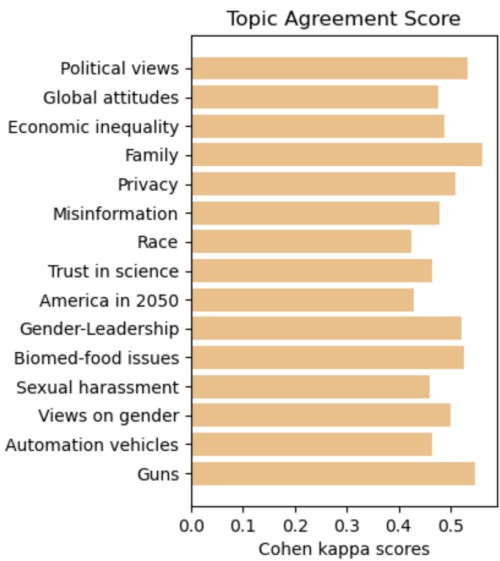

Figure 2: Topic-wise agreement score; x-axis: agreement score, y-axis: topic. This graph shows that users with similar demographics/ ideology can have different opinions (cohen kappa scores of around 0.4 show not some but not substantial correlation in opinions)

**Opinions differ despite same ideology**   To analyze the correlation between user opinions and their ideology, we extract user pairs that two users who answered at least more than 10 common questions and compare their opinions and political ideologies. Table 1 shows the percentage of user pairs sharing similar opinions, where 70% of opinions are matched between two users, and the percentages of the same ideologies and different ideologies within those user pairs. We observe that even though the users have similar opinions, around 80% of the user pairs have different ideologies. In contrast, we observe the percentage of sharing similar opinions among the users having similar ideologies is relatively higher than the percentage of sharing similar ideologies among the users having similar opinions in Appendix A. This implies that while having similar opinions does not necessarily imply shared ideologies among users, the presence of similar ideologies may suggest that users are more likely to have similar opinions. We particularly notice this phenomenon on the Guns and Family topics,

as highlighted in Table 1. While the percentage of user pairs with shared opinions is higher compared to other topics, the percentage of user pairs with differing ideologies within these pairs is notably higher than the percentage of user pairs with similar ideologies.

Based on the insights derived above, we incorporate them in our modeling approaches and analyze if these translate to the predictive performance of a model when used to predict user opinions as collected from the surveys.

## 3 Aligning LLMs with persona

In this section, we detail our task, possible modeling approaches, and evaluation protocols in Section 3.1 and discuss how to select the most relevant past opinions of a user in Section 3.3.

### 3.1 Setup

**Task**   We use LLMs to model a user; however, to concretely measure the performance, we use a simple question-answering (QA) setup. For our QA task, we use existing questions from the surveys and try to predict the choice from multiple-choice originally given to the subjects.   We use a prompting-based zero-shot approach to perform the multiple-choice QA. We experiment with gpt-4 (GPT-4), text-davinci-003 (GPT-3), gpt-3.5-turbo (GPT-3.5), Vicuna-13b, LLaMA-7b as the LLMs. Vicuna-13b (Chiang et al., 2023) is a large language model which was built upon LLaMA-13b (Touvron et al., 2023) and LLaMA-7b/13b is a transformer-based language model trained on trillions of tokens exclusively sourced from publicly available data. Vicuna-13b performs on par with ChatGPT (Chiang et al., 2023).

**Modeling Approaches**   We sample 100 users per topic. 20% of implicit questions belonging to the specific user are used as the user's implicit persona, and the rest are used to test the model's personalization ability. We have the following variants of our model where the model is gradually exposed to different levels of user information: demographic, ideological information, and user past opinions. Here is a rough sketch of what a prompt would contain for each modeling variation:

- no persona: this is a case where default LLM opinion is evaluated w.r.to the individual's opinion (Santurkar et al., 2023).

| | Guns | Auto | Gender | Sex. harass. | Biomed-food | Leadership | 2050 US | Trust-Science |
|---|---|---|---|---|---|---|---|---|
| Similar op. user pair | 45 | 13 | 30 | 12 | 11 | 37 | 23 | 21 |
| Similar op. & ideol. | 19 | 18 | 21 | 30 | 19 | 24 | 20 | 20 |
| Similar op. & diff. ideol. | 81 | 82 | 79 | 70 | 81 | 76 | 80 | 80 |
| | Race | Misinfo. | Privacy | Family | Econ. Inequal. | Global Attitudes | Politics | |
| Similar op. user pair | 12 | 29 | 21 | 43 | 25 | 24 | 16 | |
| Similar op. & ideol. | 30 | 20 | 17 | 19 | 25 | 33 | 40 | |
| Similar op. & diff. ideol. | 70 | 80 | 83 | 81 | 75 | 67 | 60 | |

Table 1: Percentage of user pairs sharing similar opinions (similar op. user pair) and the percentages of similar ideologies (similar op. & ideol.) and different ideologies (similar op. & diff. ideol.) within user pairs sharing similar opinions. (Auto: Automation, Sex. harass.: Sexual harassment, Biomed-food: Biomedical food, Misinfo.: Misinformation, Econ. Inequal.: Economic Inequality, Politics: Political Views)

- ideology: here, we observe if ideological inclinations from the user help the model to align better to them (Santurkar et al., 2023).
- ideology + demographics: here, we observe if both demographic information and ideological inclinations from the user help the model to align better with them (Santurkar et al., 2023).
- ideology + opinions: we combine ideological inclinations and opinions and measure if these help the model to align better with an individual.
- demographic + ideology + opinions: when we combine all possible personal information, i.e., demographic, ideology, and opinions, and measure if these help the model to align better with an individual. See Figure 5 for the complete prompt.

### 3.2 Evaluation Metric

For accuracy evaluation, we utilize two types of accuracy measures, underline overall accuracy and collapsed accuracy. For overall accuracy, we simply calculate the accuracy of the predicted answer choice with respect to the gold answer choice from the dataset. We also present collapsed accuracy because most answer choices in the opinion QA dataset have around 3 to 4 classes. In cases where there are more than 4 classes, it is possible to further group the classes into super classes without losing substantial finer information. For example, the following answer choices: [Very likely, Somewhat likely, Not too likely, Not at all likely], can be grouped into [Likely, Unlikely]. We consolidate such answer choices into two classes, referred to as collapsed accuracy, and present the results accordingly.

We follow Santurkar et al. (2023) for the opinion alignment score evaluation. For the alignment scores, we calculate the difference between user and model opinion distributions. The difference is calculated using *1-Wassaerstein distance* ($\mathcal{WD}$), which measures the minimum cost for transforming one distribution to the other:

$$\mathcal{A}(\mathcal{D}_m, \mathcal{D}_h; Q) = \frac{1}{Q} \sum_{q \in Q} 1 - \frac{\mathcal{WD}(\mathcal{D}_m(q), \mathcal{D}_h(q))}{N - 1}$$

where $N$ is the number of answer choices excluding refusal, and the normalization factor $N - 1$ is the maximum $\mathcal{WD}$ between two answer choice distributions. The final score can be interpreted as how well the model distribution aligns with the human distribution.

### 3.3 LLM mimicking a person with ideologies, demographic, and opinions

Our main goal is to use different components of a user's persona (demographics, ideology, opinions) to align an LLM with an individual. Specifically, by having two experiments, one with past opinions+ideology and the other with past opinions+ideology+demographics, we aim to analyze the role of demographics when predicting user responses. In addition, we hypothesize that giving users' past opinions may offer useful insights into their perspective (followed from Table 1), and LLM can benefit from that information when predicting the future answer for the specific user. When adding the user's past opinions, we compare the model with all opinions (maximum 16) to the model with top-$k$ opinions ($k$ is a hyperparameter $\in 3, 5, 8$). The top-$k$ opinions are obtained by comparing the embedding similarity between the user's previous opinions and the question at hand, where we employ text-embedding-ada-002 to obtain the embeddings. We hypothesize that all opinions

| Model | Exact match | | | | | Collapsed match | | | | |
|---|---|---|---|---|---|---|---|---|---|---|
| | L-7b | V-13b | GPT-3.5 | GPT-3 | GPT-4 | L-7b | V-13b | GPT-3.5 | GPT-3 | GPT-4 |
| No Persona | 0.33 | 0.36 | 0.37 | 0.43 | 0.53 | 0.60 | 0.62 | 0.63 | 0.62 | 0.68 |
| Demo.+Ideo. | 0.35 | 0.39 | 0.47 | 0.47 | 0.54 | 0.62 | 0.62 | 0.66 | 0.65 | 0.70 |
| Demo.+Ideo.+Opinion$_{all}$ | **0.37** | 0.41 | 0.50 | 0.51 | **0.58** | 0.61 | 0.62 | 0.69 | 0.69 | 0.73 |
| Opinion$_{top3}$ | 0.35 | 0.42 | 0.50 | 0.51 | 0.55 | 0.61 | 0.62 | 0.67 | 0.67 | 0.71 |
| Opinion$_{top8}$ | 0.36 | 0.42 | 0.50 | 0.52 | 0.56 | **0.63** | 0.63 | 0.68 | 0.68 | 0.72 |
| Ideo.+Opinion$_{top8}$ | 0.36 | **0.43** | **0.51** | 0.53 | 0.57 | 0.62 | **0.64** | 0.69 | 0.69 | 0.73 |
| Demo.+Opinion$_{top8}$ | **0.37** | **0.43** | 0.50 | 0.53 | 0.57 | 0.61 | 0.63 | 0.69 | 0.69 | 0.73 |
| Demo.+Ideo.+Opinion$_{top3}$ | 0.35 | 0.42 | **0.51** | 0.53 | **0.58** | 0.61 | 0.63 | **0.70** | 0.69 | 0.73 |
| Demo.+Ideo.+Opinion$_{top8}$ | **0.37** | **0.43** | **0.51** | **0.54** | **0.58** | 0.61 | 0.63 | **0.70** | **0.70** | **0.74** |

Table 2: Overall QA accuracy. For statistical significance, all models scored $\pm 0.01$, which are computed using Wilson score intervals for $\alpha= 99\%$. GPT-4 was tested over 50% of the testset due to cost, but we found the performance with GPT-3 with the subset remains similar to the original testset by having $< 0.004$ standard deviation. (L-7b: LLaMA-7b, V-13b: Vicuna-13b)

may incorporate some unrelated viewpoints to answer the question, and hence offering more pertinent opinions would enhance the model's ability to accurately anticipate its future response for the user. We show a complete prompt that uses all available past information of individuals to predict their future opinions and all prompts for other modeling approaches in Appendix C.

## 4 Results and Analysis

Here, we first analyze our model variants (Section 4.1) to validate hypotheses that we gather from analyzing the dataset (in Section 2.4). We also provide our model's performance when we use similar modeling setup to predict group-level opinions in Sections 4.2 and 4.3.

### 4.1 LLM for an individual

Here we discuss results of using an LLM to model an individual in the light of the evaluation metrics described in Section 4.1.

**Exact match vs. Collapsed match** The accuracy with the exact match and with the collapsed match in Table 2 shows a similar trend for the performance of our model variants. This suggests that leveraging implicit opinions enables the model to align with the correct range of answer choices, even though it does not precisely predict the exact same answer as the user's choice.

**Overall Accuracy** Table 2 presents overall QA accuracy with exact match and collapsed match for answer choices. In most cases, utilizing demographic+ideology+top-8 opinions performs the best. Moreover, adding demographic and ideology information outperforms the model without

any persona, indicating that some questions might be highly correlated with the user's demographics, and LLM is able to make a guess with the demographic information. Incorporating the user's previous opinions, up to 16 in total, along with demographic information, substantially enhances the performance in both overall and collapsed accuracy across all models. This implies that users' past opinions are indeed important to make correct predictions.

Interestingly, utilizing the top-$k$ most relevant previous opinions does not yield a significant increase in collapsed accuracy. However, it does improve the exact match accuracy by up to 3 points with GPT-3 when using both demographics and ideology along with the user's previous opinions. This implies that having top-$k$ most relevant past opinions can help the model pinpoint more accurate answers, and providing the user's past opinions is already pushing the model to be in the correct range of the answer choices. We noticed that utilizing the top-3 opinions yields similar performance to using the top-8 opinions, indicating that a few of the most relevant opinions carry the most performance improvement of the model.

Moreover, simply using the top 3 most relevant opinions performs on par with the model with user demographic, ideology, and user's past 16 random opinions. This confirms again that utilizing the most relevant opinions as feedback is essential to get personalized answers from LLM. Lastly, providing additional demographic information with ideology slightly improves the model performance, implying that the demographic information may contribute valuable insights to the model to a certain degree.

| Model | GPT-3 | GPT-3+CoT |
|---|---|---|
| Opinion$_{top8}$ | 0.52 | 0.51 |
| Ideo.+Opinion$_{top8}$ | 0.53 | 0.52 |
| Demo.+Opinion$_{top8}$ | 0.53 | 0.52 |
| Demo.+Ideo.+Opinion$_{top8}$ | 0.54 | 0.53 |

Table 3: QA accuracy comparison between GPT-3 and GPT-3 with Chain of Thought style prompt.

**Comparison between LLMs**  GPT-4 produces a similar performance when using all opinions and when using top-$k$ relevant opinions. This implies that GPT-4 itself might be able to identify the most relevant user's previous opinions.

We find that the LLaMA-7b model does not understand what to produce when opinions are added to the prompt resulting in 18% cases where it does not yield any answer. Likewise, GPT-3.5 tends to produce no answer for questions that are about personal opinions (e.g. GPT-3.5 with top-3 opinions produces 59% no-answers). When we prompted the model to generate an answer without an explanation by changing the suffix of the prompt: "Answer choice:" → "Answer choice without explanation:", and substantially fewer (less than 1%) cases have no-answers.

The trend of the performance where we add top-$k$ opinions remains the same for both GPT-3 and GPT-3.5. There is no clear winner between different versions of GPT as also found in the other literature (Fu et al., 2023).

**Chain of Thought (CoT) Prompting**  CoT (Wei et al., 2023) has shown that encouraging LLMs to explain their reasoning step-by-step improves the model performance. To see whether CoT also benefits in personalizing answers, we test the GPT-3 model with CoT-style prompting by changing the suffix of our prompt from "Answer choice:" to "Let's think step by step and choose one answer choice:" and present the results in Table 3. We discover that using the CoT prompt consistently decreases the performance by 1 point. This is because the model attempts to provide the reason for the potentially irrelevant information presented in the prompt, suggesting its inability of selecting the most relevant information about the answer. E.g., in Figure 4, the model generates reasoning exclusively based on demographic and ideology information but ignores user opinions.

**Opinion Alignment Scores**  Among high-performing models, since only GPT-3 provides prediction probabilities, we compare opinion

| Model | Opinion Alignment Score |
|---|---|
| No Persona | 0.670 |
| Demo.+Ideo. | 0.763 |
| Demo.+Ideo.+Opinion$_{all}$ | 0.780 |
| Opinion$_{top3}$ | 0.777 |
| Opinion$_{top8}$ | 0.779 |
| Ideo.+Opinion$_{top8}$ | 0.793 |
| Demo.+Opinion$_{top8}$ | 0.789 |
| Demo.+Ideo.+Opinion$_{top3}$ | **0.796** |
| Demo.+Ideo.+Opinion$_{top8}$ | **0.795** |

Table 4: Opinion alignment scores (§3.2) with GPT-3 (text-davinci-003).

alignment scores (defined in §3.2) with various setups using GPT-3 model in Table 4. Overall scores exhibit a similar trend as seen in Table 2, in which utilizing top-$k$ most relevant opinions outperforms the model that uses all opinions. This is consistent with our finding that having top-$k$ most relevant past opinions helps the model find more accurate answers for the user.

**Topic-wise Accuracy**  Figure 3 demonstrates the model's accuracy across different topics with various setups, measured by the exact match for answer choices. The model with demographic and implicit opinions particularly achieves higher scores on the Biomedical-food and Guns topics, implying that these two topics may lead the users to have similar opinions of each other. In contrast, the model exhibits slightly decreased performance when incorporating implicit opinions on topic Automation. This suggests that the LLM can make accurate predictions up to some extent based on user demographic and ideology information. However, incorporating implicit opinions, which may include viewpoints not aligned with users' demographic or ideologies, can potentially confuse the model in its prediction process.

**Common Errors**  In Table 5, we manually analyzed 30 randomly sampled opinions where GPT-3 produces correct answers with demographic+ideology information but incorrect answers when opinions are added. The model mostly confuses when there is a high overlap between user opinions and some answer (possibly wrong) choices. For a question: "How well do the following phrases describe you?" and a correct answer: "Describes me well," the model often predicts correctly purely based on demographic information (e.g., Black). However, when the user's past opinions were added and since it includes the phrase "supporter of rights for LGBT people do not de-

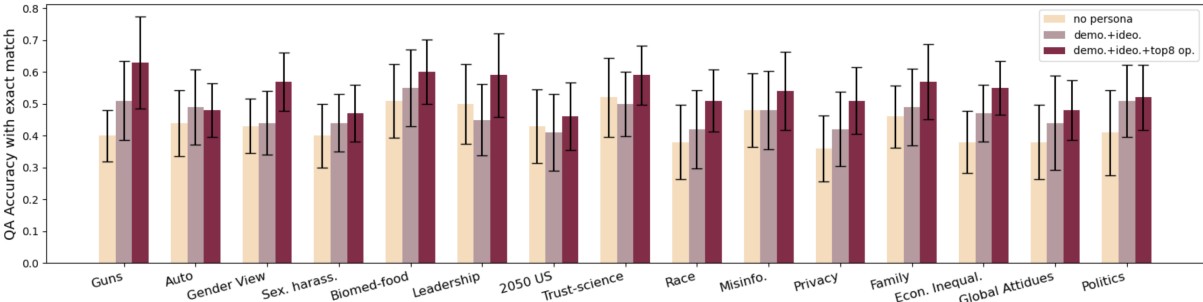

Figure 3: Overall topic-wise accuracy based on the exact match for answer choices. (demo.: demographic, ideo.: ideology, op.: opinions). GPT-3 model is used for the experiment. See Appendix B for the scores.

| Opinion contains | Percentage (%) |
|---|---|
| Word-overlap with answer choice | 36 |
| Irrelevant information | 30 |
| Relevant information | 30 |

Table 5: Types of common errors with their percentages when prompting with opinions confuses the model.

scribe me well," it predicts a wrong answer choice: "does not describe me well" just because the direct phrase overlap. While models often ignore additional relevant information and commit errors, it is also sensitive to irrelevant opinions (even in top-$k$) when additionally added with demographic information. For e.g., while predicting user opinion on the U.S. remaining as a world superpower in 2050, the user's past opinion about violence against Jews (retrieved in top-8) confuses the model leading to an incorrect prediction. This also explains why just adding top-3 opinions often archives the highest accuracy instead of adding all opinions. See Appendix D for examples of all types of errors.

## 4.2 LLM with majority answer choices

Additionally, we also wanted to understand if similar performances can be achieved if we model an individual as a member of a (sub-)population, mirroring (Santurkar et al., 2023). While it is true that opinions are necessary to predict individual-level preferences, here we investigate if demographic information is sufficient for predicting population-level opinions or not. For this, we first merge our QA data points using a particular ideological group value (e.g., democrat) and obtain the answer choice that is chosen by most of the group members (i.e., a majority vote) and treat that answer as the gold answer for the question while calculating the accuracy (Kim and Lee, 2023).

We prompt our model to predict an answer given a question assuming the role of a group representa-

> Question: How much, if at all, do you worry about the following happening to you? Being the victim of a terrorist attack
>
> Choices: Worry a lot, Worry a little, Do not worry at all
>
> Opinions: I never used a gun to threaten or intimidate anyone in family.
> There were no guns in my household when I was growing up.
>
> Demographics and Ideology:
> Political party: Republican, Political ideology: Conservative, Region: South, Marital status: Married, Education: College graduate/some postgrad
>
> **CoT reasoning:**
> Step 1: The person is conservative and Republican, so they may be more likely to worry about a terrorist attack.
> Step 2: The person is married and has a college degree, so they may be more aware of the potential risks of a terrorist attack.
> Step 3: The person is in the South, so they may be more likely to worry about a terrorist attack due to the region's history of terrorism.
> Conclusion: The person is likely to select answer choice A: Worry a lot.
>
> User-answer (expected): Do not worry at all
> Demo.+ideo.+opinions with CoT: Worry a lot ✗
> Demo.+ideo.+opinions w/o CoT: Do not worry at all ✓

Figure 4: An example of incorrect CoT reasoning confusing the model. As we see in CoT reasoning, the model reasons about demographic information, not opinions provided.

tive, i.e., a person having a majority vote answers belonging to a specific group (See Appendix C for the prompt). We see that the LLM is good at predicting the answer given by the majority of the group member belonging to a certain ideology, suggesting that LLMs are good at modeling a representative individual of a sub-population (e.g., all democrats). The overall performance without ideology information is 0.597 (with exact answer choice

|                  | Exact match | Collapsed match |
|------------------|-------------|-----------------|
| Majority answer  | 0.597       | 0.674           |
| Independent      | 0.546       | 0.674           |
| Democrat         | 0.578       | 0.665           |
| Republican       | 0.523       | 0.639           |

Table 6: Performance with LLM with ideology information. GPT-3 model is used for the experiment.

match) and 0.674 (with collapsed answer choice match), as presented in Table 6. This also indicated that the default opinions from the LLMs are somewhat aligned with the majority opinions seen at a population level.

### 4.3 LLM as a person with an ideology

Next, we add the ideological information to see if this additional information can help the LLM perform better to model a user belonging to a group that believes in a specific ideology (e.g., conservative) (See Appendix C for the prompt). We find that the LLM is moderately good at modeling a user with group-level information to predict the group-level majority opinion. This indicates that the additional ideological information is not particularly helpful. The average performance with ideology information is 0.549 (with exact answer choice match) and 0.659 (with collapsed answer choice match), as shown in Table 6. We see a similar trend in results for modeling an individual with their demographics and/or ideology and/or past opinions since an individual's opinion does not align with the group's majority opinion that the person belongs to.

### 5 Related work

**Personalization**   Past works that focused on modeling individual users were from the pre-LLMs era and mainly hail from the recommender systems literature (Gao et al., 2023b; He et al., 2017; Li et al., 2021; Majumder et al., 2019). However, these systems were trained on domain-specific annotated datasets or using latent information about the users (e.g. their previously written reviews). The recent LLMs have seen less content from the long tail of user groups during their pre-training phase, and there has been a lack of large-scale datasets of individual opinions until recently (Santurkar et al., 2023). Thus it remains an open problem whether LLMs can be aligned effectively with individual user persona and how different user information

(e.g., demographic traits vs. past opinions) influences how well an LLM can model individual's opinions. For a comprehensive comparison among all previous work, see Table 9.

**Role of demographics and ideology**   There have been several studies investigating the correlation between ideological attitudes and psychological traits (Zmigrod et al., 2021; Crockett and Wallendorf, 2004; Chan and Palmeira, 2021). Crockett and Wallendorf (2004) found that normative political ideology is central to understanding shopping as a manifestation of social and political connections. Chan and Palmeira (2021) found that the cognitive decision-making strategies of individuals reflect their ideological attitudes. Differently, we show that ideology is not the only important factor in predicting the user's opinion using an LLM.

**LLMs with retrieval-based approach**   Extensive prior work has used retrievals from a text corpus to aid QA (Madaan et al., 2022; Pan et al., 2019), or retrievals of prior QA pairs for nearest-neighbor QA (Khandelwal et al., 2020). Madaan et al. (2022) uses a memory of user opinions to retrieve past relevant data points for the prompt. Khandelwal et al. (2020) showed the effectiveness of the nearest neighbor search for language modeling by extending a pre-trained language model (LM). Differently from work on LLMs and group-level personalization, we show that LLMs can be tuned for individual users with their opinions.

### 6 Conclusion and Outlook

This paper offers a new insight that aligning LLMs to users is best done by modeling user demographics, ideologies, and the most relevant past opinions. Large-scale experiments on PEW surveys present in the OpinionQA dataset show an approximately 7% absolute QA accuracy over strong demography-based baselines. We proactively offer suggestions to avoid personalized LLMs from becoming echo chambers (see Ethics Statement). An exciting future direction is to continuously store user opinions and grow the memory of opinions.

An aligned LLM offers the benefit to offer personalized perspectives that align with a user's values and cultural beliefs. However, there exist circumstances when LLMs can become an amplifier for unethical and biased views. Our work lays the foundation for a robust LLM alignment approach. By using memory-based personalization

and recording interactions saved in a growing memory, the model can inform future instances of the *most relevant* past opinions.

## Limitations

**Non-subjective questions.** For non-subjective questions, such as "How many years have you lived in ...", it might not be necessary to use past opinions (see error analysis).

**Lack of user information.** In this work, we focus on all past opinions on the same topic due to a lack of availability of user information in the dataset. More insights can be derived by investigating users' opinions on multiple different related topics. We leave it as our future work.

**Lack of temporal information.** Another missing aspect is time. User opinions change over time, but the dataset available to us does not contain timestamps. It would be interesting to see how conflicting user opinions can be modeled, perhaps by biasing toward the most recent opinion.

## Ethics Statement

**Data** The dataset used in our work, OpinionQA (Santurkar et al., 2023) is publicly available. The dataset includes subjective opinions from humans and may contain offensive content to some people.

**Models** The large language models we used for the experiments are trained on a large-scale web corpus and some of them utilize human feedback. This may also bring some bias when predicting user answers. With an aligned LLM, users can select information that adheres to their system of beliefs and to amplify potentially biased and unethical views. Such an echo chamber (Del Vicario et al., 2016) can eventually cause harm by reinforcing undesirable or polarized a user's views.

A viable mitigation is to show user demography or ideology group answers in addition to the personalized answer (e.g., showing how an average Democrat with similar demographics would think on this topic and why). Further, past opinions can be used to ground an explanation (e.g., the current personalized answer is influenced by a user's specific past opinion), thus offering an opportunity for the user to introspect their past opinions.

## Acknowledgements

We thank the members of the Aristo team at AI2 and Kurt Gray for their insightful feedback on this work. EH was funded, in part, by the Vector Institute for AI, Canada CIFAR AI Chairs program, an NSERC discovery grant, and a research gift from AI2. BPM was funded, in part, by an Adobe Research Fellowship.

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

## A  Similar ideologies and different opinions

We show the percentage of user pairs having similar ideologies and the percentages of user pairs having similar opinions and different opinions within the user pairs sharing similar ideologies in Table 7.

## B  Topicwise QA Accuracy

We show an overall topic-wise accuracy based on the exact match for answer choice in Table 8.

## C  Prompt

We provide a comprehensive display of all prompts used in the models incorporating user demographics, ideology, and opinions, which were employed for individual-user level tests in Figure 5, 6 and 7. Additionally, we present the prompts utilized for experiments conducted at the group-level tests in Figure 8 and 9.

## D  Error Examples

We give additional error examples in Figure 10, 11, and 12, where opinions either confuse the model with relevant information or provide no useful information.

Figure 10 shows the second common error, which is when Relevant information in opinions confuses the model. For example, when the question is "Is what you know about dietitians because you have heard or read about this in the news" and the user's past opinions contain "I did not learn about dietitians in my job.". In this case, while the user's answer to the question is "Yes, have heard or read about this in the news", the model produces "No, have not heard or read about this in the news" because of the opinions provided.

The third most common error is with Irrelevant information as shown in Figure11. Some opinions do not contain useful information at all to reason for the correct answer. Lastly, there are some questions that opinions are not important to predict the answer (e.g. "About how many years have you lived in your local community?") as presented. In this case, the model may make an arbitrary prediction.

## E  Additional related work comparison

We show additional related works that can be compared to our paper in Table 9.

```
A person can be described as follows:

Age: 30-49
Income: 75,000-100,000
Political ideology: Conservative
Political party: Republican
Religion: Roman Catholic
...

The person has the following opinions on Guns.

Opinions:
1.  The most important reason why I own a
gun is for sport shooting, including target
shooting and trap and skeet.
2. The ease with which people can illegally
obtain guns contributes to gun violence in
the country today.
...

Based on the above list of opinions and the
demographic information, which answer choice
will this person select for the question:

Question: Thinking about gun owners who do
not have children in their home how important
do you think it is for them to: Take gun
safety courses

Answer choices:
A. Essential
B. Important but not essential
C. Not important
D. Should not be done

Answer:
```

Figure 5: Prompt using demographics, ideology, and GPT embeddings based top-$k$ past opinions to predict the answer to a question.

```
A person has the following opinions on Guns.

Opinions:
1. < opinion1 >
2. < opinion2 >
...

Based on the above list of opinions, which
answer choice will this person select for the
question:

Question: < question >

Answer choices:
A.< choice1 >
B.< choice2 >
C.< choice3 >
...

Answer:
```

Figure 6: prompt for implicit-only model

| Topic | similar ideol. user pair (%) | similar ideol. & op. (%) | similar ideol.-diff. op. (%) |
|---|---|---|---|
| Guns | 16.05 | 52.67 | 47.33 |
| Automation | 15.99 | 15.26 | 84.74 |
| Views on gender | 16.26 | 39.58 | 60.42 |
| Sexual harassment | 16.95 | 22.14 | 77.86 |
| Biomedical, food | 15.69 | 13.44 | 86.56 |
| Gender, Leadership | 17.52 | 50.25 | 49.75 |
| America in 2050 | 15.34 | 30.21 | 69.79 |
| Trust in Science | 15.53 | 26.27 | 73.73 |
| Race | 16.32 | 21.05 | 78.95 |
| Misinformation | 16.22 | 34.83 | 65.17 |
| Privacy, Surveillance | 16.13 | 22.14 | 77.86 |
| Family, Relationships | 17.01 | 48.88 | 51.12 |
| Economic inequality | 16.24 | 39.25 | 60.75 |
| Global attitudes | 16.75 | 47.78 | 52.22 |
| Political views | 16.65 | 37.59 | 62.41 |

Table 7: Percentage of user pairs sharing similar ideologies (similar ideol. user pair) and the percentages of similar opinions (similar ideol. & op.) and different opinions (similar ideol.-diff. op.) within user pairs sharing similar ideologies.

| | Accuracy with exact match | | |
|---|---|---|---|
| | No Persona | Demo. + Ideo. | Demo. + Ideo.+ Opinion$_{top8}$ |
| Guns | $0.40 \pm 0.08$ | $0.51 \pm 0.13$ | $\mathbf{0.63} \pm 0.14$ |
| Automation | $0.44 \pm 0.10$ | $\mathbf{0.49} \pm 0.12$ | $0.48 \pm 0.09$ |
| Views on gender | $0.43 \pm 0.09$ | $0.44 \pm 0.10$ | $\mathbf{0.57} \pm 0.09$ |
| Sexual harassment | $0.40 \pm 0.10$ | $0.44 \pm 0.09$ | $\mathbf{0.47} \pm 0.09$ |
| Biomedical, food | $0.51 \pm 0.12$ | $0.55 \pm 0.12$ | $\mathbf{0.60} \pm 0.10$ |
| Gender, Leadership | $0.50 \pm 0.13$ | $0.45 \pm 0.11$ | $\mathbf{0.59} \pm 0.13$ |
| America in 2050 | $0.43 \pm 0.12$ | $0.41 \pm 0.12$ | $\mathbf{0.46} \pm 0.11$ |
| Trust in science | $0.52 \pm 0.12$ | $0.50 \pm 0.10$ | $\mathbf{0.59} \pm 0.09$ |
| Race | $0.38 \pm 0.12$ | $0.42 \pm 0.12$ | $\mathbf{0.51} \pm 0.10$ |
| Misinformation | $0.48 \pm 0.12$ | $0.48 \pm 0.12$ | $\mathbf{0.54} \pm 0.12$ |
| Privacy, Surveillance | $0.36 \pm 0.10$ | $0.42 \pm 0.12$ | $\mathbf{0.51} \pm 0.11$ |
| Family, Relationships | $0.46 \pm 0.10$ | $0.49 \pm 0.12$ | $\mathbf{0.57} \pm 0.12$ |
| Economic inequality | $0.38 \pm 0.10$ | $0.47 \pm 0.09$ | $\mathbf{0.55} \pm 0.08$ |
| Global attitudes | $0.38 \pm 0.12$ | $0.44 \pm 0.15$ | $\mathbf{0.48} \pm 0.10$ |
| Political views | $0.41 \pm 0.13$ | $0.51 \pm 0.11$ | $\mathbf{0.52} \pm 0.10$ |

Table 8: Overall topic-wise accuracy based on exact match and collapsed match for answer choices. (demo.: demographic, ideo.: ideology, op.: opinions). GPT-3 model is used for the experiment.

| | User-profile explicitly observed | Modeling individuals | Supervision-free |
|---|---|---|---|
| **Personalized generation**: OpinionQA (Santurkar et al., 2023), RecipeGen (Majumder et al., 2019), LAMP (Salemi et al., 2023) | ✗ | ✗ or ✓ (mostly group) | ✗ or ✓ |
| **Recommender Systems**: ChatRec (Gao et al., 2023b), Collaborative Filtering (He et al., 2017), BotPlay (Li et al., 2021) | ✗ or ✓ (mostly latent) | ✓ | ✗ or ✓ (mostly supervised) |
| **Ours** | ✓ | ✓ (+ group) | ✓ |

Table 9: Placement of our work w.r.t. related work

```
A person can be described as follows:

Age: < age >
Education: < education >
...

Based on the demographic information, which
answer choice will this person select for the
question:

Question: < question >

Answer choices:
A.< choice1 >
B.< choice2 >
C.< choice3 >
...

Answer:
```

Figure 7: prompt for demographic-only model

```
Thinking    of    yourself    as    a
[republican/independent/democrat],    please
select the right choice.

< question >

Choice: [choice1, choice2, choice3]
Answer:
```

Figure 8: prompt for group-level ideology test

```
Thinking of yourself as a person, please
select the right choice.

< question >

Choice: [choice1, choice2, choice3]
Answer:
```

Figure 9: prompt for LLM majority answer test

```
Question:
Is what you know about dietitians because you
have heard or read about this in the news

Choices:
Yes, have heard or read about this in the
news
No, have not heard or read about this in the
news
Refused

Opinions:
I did not learn about dietitians in my job.
...

Demographics and Ideology:
Political party: Republican,
Race: White,
Religion: Protestant,
...

User-answer (expected): Yes, have heard or
read about this in the news

Demo.-ideology-opinions: No, have not heard
or read about this in the news ✗

Demo.-ideology: Yes, have heard or read about
this in the news ✓
```

Figure 10: An example of a relevant opinion confusing
the model.

```
Question:
How likely do you think it is that the
following will happen in the next 30 years?
China will overtake the U.S. as the world's
main superpower

Choices:
Yes, have heard or read about this in the
news
No, have not heard or read about this in the
news
Refused

Opinions: (irrelevant)
Population growth in the U.S. will be a major
problem in 2050.
It is probable that there will be increasing
violence against Jews in the U.S. in the next
30 years.
...

Demographics and Ideology:
Citizenship: Yes, Political party: Moderate,
Race: White,
Region: Midwest,
...

User-answer (expected):  Will probably not
happen

Demo.-ideology-opinions: Will probably happen
✗

Demo.-ideology: Will probably not happen ✓
```

Figure 11: An example of opinions containing no useful information to predict the answer.

```
Question:
How well, if at all, do the following words
or phrases describe you?  Supporter of the
Black Lives Matter movement

Choices:
Describes me well
Does not describe me well
Refused

Opinions:
The words or phrases "supporter of rights for
LGBT people" do not describe me well.
...

Demographics and Ideology:
Political party: Democrat,
Race: Black,
Religion: Protestant,
...

User-answer (expected): Describes me well

Demo.-ideology-opinions: Does not describe
me well ✗

Demo.-ideology: Describes me well ✓
```

Figure 12: An example of a not relevant opinion confusing the model