# OpenReview forum: "Aligning Language Models to User Opinions"
_EMNLP/2023/Conference — EMNLP 2023 Findings_

### Official Review · Reviewer_cFeH · 2023-08-05

**Soundness:** 4

**Excitement:**

2: Mediocre: This paper makes marginal contributions (vs non-contemporaneous work), so I would rather not see it in the conference.

**Paper Topic And Main Contributions:**

This study investigates the effects of user factors on LLMs. The authors conducted experiments on a recent data, OpinionQA. They found that aligning LLMs by user opinions with user demographics and ideology can boost LLM performance.

**Reasons To Accept:**

1. This is very novel study that investigates effects of user factors.
2. The analysis is solid.

**Reasons To Reject:**

I rated highly for the submission quality, BUT according to the EMNLP policy, the arxiv version (May 24) is after the anonymized date (May 23). This should be rejected before the review.

**Reproducibility:**

5: Could easily reproduce the results.

**Reviewer Confidence:**

4: Quite sure. I tried to check the important points carefully. It's unlikely, though conceivable, that I missed something that should affect my ratings.

---

> ### Author Rebuttal · Authors · 2023-08-29
>
> Thank you for taking the time to read our paper! We are happy that you appreciated our main points: 1) investigating effects of different user factors and 2) solid analysis across different LLMs. We would love to address additional questions during the discussion period if you have more questions.
>
> **arxiv version (May 24) is after the anonymized date (May 23)**
>
> We submitted the arxiv version before 11:59 pm UTC -12h (“anywhere on Earth”) on May 23rd, 2023 and the arxiv timezone does not follow AoE timezone. This is our submission time: Wed, 24 May 2023 09:11:11 UTC (AoE is UTC-12 -- and thus this is 23 May 2023, 9PM UTC -12h; which is roughly three hours before the deadline)

---

### Official Review · Reviewer_Qyo6 · 2023-08-05

**Soundness:** 3

**Excitement:**

2: Mediocre: This paper makes marginal contributions (vs non-contemporaneous work), so I would rather not see it in the conference.

**Paper Topic And Main Contributions:**

The paper proposes a method to align personalized user opinions on large language models (LLMs). The authors leverage demographic information, ideology, selected previous opinions, and chain-of-thought prompting to achieve this alignment. They conduct experiments using the OpinionQA dataset, which consists of 5K users' answers to topical questions. Various pretrained LLMs are utilized to measure the accuracies of LLMs in providing appropriate responses based on the given persona and additional information as prompts. The experimental results demonstrate that using all the suggested features outperforms models that lack these ideas.

**Questions For The Authors:**

1. Why is it necessary to align a decoder-based LLM to a user to address the opinion classification task? Can you provide insights into the performance of encoder-based LLMs, such as BERT and RoBERTa, when the primary goal is to predict user opinions based on demographic information and previous opinions? Furthermore, what other tasks can be achieved after aligning an LLM to a user? For instance, can personalized responses be generated for a given conversation using the suggested approach?

2. The claim that "human feedback is not necessarily helpful in personalizing the answers" (lines 376 ~ 378) is hard to believe, especially with only the difference between the performance of GPT-3 and GPT-3.5. Could you clarify if this is the only difference between GPT-3 and GPT-3.5? It would be more convincing to present additional experiment results with other LLMs that do use human feedback and compare them to those that do not.


**Reasons To Accept:**

- The paper introduces an interesting and well-founded prompt method for predicting personalized opinion classes.
- The research presents diverse experiment results involving different ideas and LLMs.


**Reasons To Reject:**

- The task appears to resemble a classification task, so the paper should clarify why the focus is on decoder-based LLMs.
- The claims made in the paper might be overstated, given the limited number of experimental results.


**Reproducibility:**

5: Could easily reproduce the results.

**Reviewer Confidence:**

4: Quite sure. I tried to check the important points carefully. It's unlikely, though conceivable, that I missed something that should affect my ratings.

---

> ### Author Rebuttal · Authors · 2023-08-29
>
> Thank you for taking the time to read our paper and pointing out important discussion points! We have made those more clear in the paper now. We are happy that you appreciated our main points: 1) well-founded prompts using demographics, ideology, and opinions, and 2) diverse experiments across different LLMs. We would love to address additional questions during the discussion period if anything is unclear.
>
> **What would be the performance of encoder-based LLMs, such as BERT and RoBERTa on this task?**
>
>
> Thanks for pointing it out. We ran additional experiments with BERT model, one with demographic+ideology and the other with demographic+ideology+top8 opinions. The performance is as follows:
>
> demographic+ideology: 0.508
>
> demographic+ideology+top8 opinions: 0.523
>
> The performance trends are similar to GPT3+ models. While the performance is almost comparable to GPT3, it requires substantial training with more than 10k points. Thus the main finding of our paper, in which using all information (demographic+ideology+opinion) is helpful in aligning the LLM to a user, still holds whether they are open source or closed-source. We also provide insights into how various user factors (such as demographics, opinions, and ideologies) influence the behavior of LLMs using a few shot demonstrations.
>
> Experimental setup details are: bert-base-uncased, 2 epochs, learning rate of 0.00001, batch_size of 8. Developing more advanced encoder-based model baselines for our task would be an interesting future direction.
>
> **What other tasks can be achieved after aligning an LLM to a user? (e.g. personalized response generation?)**
>
> You are exactly right. We hope to generate personalized responses by augmenting our setup with a memory of user opinions. The idea of a memory-augmented architecture has been recently explored [1] -- where the memory grows with user interactions and feedback. For a future instance, relevant past memory items are retrieved (like we show in our strongest model “top-k opinions”). This avoids repeating mistakes where a user specifies their opinions and also aligns well with users. The personalization could then align with a user’s ethical principles, moral beliefs, and cultural-specific beliefs. This is an unsolved problem yet with LLMs and we believe that we offer the first insight into why modeling such opinions could be valuable and eventually lead to personalized LLMs.
>
> [1] Memory-assisted prompt editing to improve GPT-3 after deployment, Madaan et al., EMNLP 2022 (https://arxiv.org/abs/2201.06009)
>
> **GPT3 vs GPT3.5: Clarification on the intuition that human feedback is not necessarily helpful in personalizing the answers**
>
> You are right. We checked the open ai model version we used again and we realized RLHF was applied to text-davinci-003 model. We missed this detail since the information was not explicitly mentioned on the website and based on this information we made the hypothesis; however, based on our corrected understanding, this claim is not really correct-- thanks for pointing it out. We will remove the sentence related to RLHF in one of the smaller analysis subsections in the camera-ready version.
>
> As for the performance difference between GPT and GPT turbo -- There is no clear winner between different versions of GPT as also found in the literature elsewhere (See https://arxiv.org/abs/2305.17306). On BBH(BigBench Hard): Turbo 70.1, davinci003 70.7, davinci002 73.7 -- it is thus conceivable that for our task Turbo is slightly worse than GPT3.

---

### Official Review · Reviewer_iqqj · 2023-08-05

**Soundness:** 4

**Excitement:**

4: Strong: This paper deepens the understanding of some phenomenon or lowers the barriers to an existing research direction.

**Paper Topic And Main Contributions:**

The paper addresses the problem of aligning large language models (LLMs) with individual users' personalities and opinions. It aims to personalize LLMs to better understand and respond to individual users' preferences, instead of aligning with broad demographic or ideological groups. This paper also highlights the importance of considering users' past opinions as an essential ingredient in aligning language models. By using relevant past opinions, the model can better understand and respond to a user's unique perspective, leading to more accurate and personalized answers.

**Questions For The Authors:**

QA: Have you considered other factors or sources of user-specific information that could further enhance model alignment?

**Reasons To Accept:**

- The paper addresses a critical and novel question in the field of natural language processing (NLP) about how to best align language models with individual users, moving away from the traditional focus on demographic or ideological groups. This question is highly relevant given the increasing use of large language models in real-world applications.

- The paper conducts a thorough comparative analysis of different LLM variants, including those with demographic information, ideology, and past opinions. This analysis provides valuable insights into the impact of different factors on model alignment and performance.

- The paper provides detailed experimental results and analysis, allowing other researchers to reproduce and validate the findings. This can foster further research and comparisons of different alignment techniques in the NLP community.

**Reasons To Reject:**

- The paper heavily relies on the OpinionQA dataset, which may not fully capture the diversity of user opinions and demographics. The generalizability of the findings could be limited to the specific dataset used.

**Reproducibility:**

3: Could reproduce the results with some difficulty. The settings of parameters are underspecified or subjectively determined; the training/evaluation data are not widely available.

**Reviewer Confidence:**

3: Pretty sure, but there's a chance I missed something. Although I have a good feel for this area in general, I did not carefully check the paper's details, e.g., the math, experimental design, or novelty.

---

> ### Author Rebuttal · Authors · 2023-08-29
>
> Thank you for taking the time to review our paper and recognizing our contributions! We are happy that you appreciated our main points: 1) aligning language models with individual users by using demographics, ideology, and opinions, 2) thorough comparative analysis across different LLM variants, and 3) valuable insights into the impact of different factors on model alignment and performance.
> We think that all your questions are addressable within this discussion period. We would love to address additional questions during the discussion period if anything is unclear.
>
>
> **Limited generalizability of the findings with the specific dataset used**
>
> Thank you for raising this concern. OpinionQA covers 60 US demographic groups over 15 topics ranging from automation to politics. While it covers a large amount of the population, we agree that the insights may predominantly apply to the U.S. population. We hope there are more datasets across demographics that the community should work on and that can further validate the generalization of our results.
>
> **Other sources of user-specific information that could further enhance model alignment**
>
>
> That’s a good point and we exploited all the user information present in the opinionQA dataset. While demographics and ideologies represent conventional features in psychology for modeling, we additionally use the dataset for extra information in the form of opinions and show that this is a very important signal. More detailed demographic features such as those present in small-scale user studies in psychology and cognitive science are helpful but need to be carefully engineered. However, we demonstrate that opinions which have been less exploited as a signal are helpful in predicting user responses and are a more powerful and important signal than demographics. On the other hand, synthetically adding auxiliary user information can lead to stereotyping and cannot be used as gold data.

---

### Meta-Review · Area_Chair_tZLP · 2023-09-15

**Recommendation:** 3

**Metareview:**

This paper proposes an approach to align large language models (LLMs) to individual users by modelling the user's demographics, ideology, and past opinions. The key idea is to go beyond just using demographics or ideology and to also leverage the user's previous stance on issues as a richer signal for personalization. The authors conduct experiments on the OpinionQA dataset using different LLM variants and show that modelling all factors - demographics, ideology, and past opinions - leads to accuracy gains in predicting user responses.

The reviewers raise important points about the generalizability of the findings beyond this specific dataset and whether other user factors could further enhance personalization. In their rebuttal, the authors provide satisfactory responses, conducting additional experiments with BERT to demonstrate consistent trends. The authors also thoughtfully discuss how their approach provides a scaffolding for generating personalized responses in future work, albeit requiring large-scale user studies.

Reviewer 2 suggests encoder-based models may be sufficient for the task framed as classification, while the authors clarify the benefits of studying decoder models and few-shot prompting. Reviewer 3 recommends rejection based on anonymization rules, which is an important issue that needs resolution.

Overall, this appears to be a novel contribution demonstrating the utility of modelling opinions in addition to standard user factors for alignment. Furthermore, the paper tackles an important problem of personalizing large language models (LLMs) to individual users. Aligning LLMs to unique user perspectives, beyond broad demographic or ideological groups, can enable more accurate and natural conversational AI.

I think the core idea is sound and I agree with the reviewers that the experiments are reproducible, and results are likely generalizable beyond this dataset based on additional analyses.  The core contribution is demonstrating that modelling a user's past opinions is a critical signal, on par with or even more informative than standard factors like demographics and ideologies. This insight could reframe debates in social sciences on whether ideologies alone capture individual variance. The authors thoroughly analyze different prompting combinations on multiple LLMs using the OpinionQA dataset. The results clearly show gains from adding past opinions to user profiles, with the full model reaching up to 7% higher accuracy. Additional BERT experiments confirm the consistency of these trends.

I think the paper is technically sound, with reproducible experiments and sufficient analysis, especially since the framing and motivation situate the work well within active discussions on model personalization in the NLP community.

In conclusion, the work provides valuable insights into modelling different user factors, laying the groundwork for personalized applications.

---

### Decision · Program_Chairs · 2023-10-07

**Decision:**

Accept-Findings

**Comment:**

This paper proposes an approach to align large language models (LLMs) to individual users by modelling the user's demographics, ideology, and past opinions. The key idea is to go beyond just using demographics or ideology and to also leverage the user's previous stance on issues as a richer signal for personalization. The authors conduct experiments on the OpinionQA dataset using different LLM variants and show that modelling all factors - demographics, ideology, and past opinions - leads to accuracy gains in predicting user responses.

The reviewers raise important points about the generalizability of the findings beyond this specific dataset and whether other user factors could further enhance personalization. In their rebuttal, the authors provide satisfactory responses, conducting additional experiments with BERT to demonstrate consistent trends. The authors also thoughtfully discuss how their approach provides a scaffolding for generating personalized responses in future work, albeit requiring large-scale user studies.

Reviewer 2 suggests encoder-based models may be sufficient for the task framed as classification, while the authors clarify the benefits of studying decoder models and few-shot prompting. Reviewer 3 recommends rejection based on anonymization rules, which is an important issue that needs resolution.

Overall, this appears to be a novel contribution demonstrating the utility of modelling opinions in addition to standard user factors for alignment. Furthermore, the paper tackles an important problem of personalizing large language models (LLMs) to individual users. Aligning LLMs to unique user perspectives, beyond broad demographic or ideological groups, can enable more accurate and natural conversational AI.

I think the core idea is sound and I agree with the reviewers that the experiments are reproducible, and results are likely generalizable beyond this dataset based on additional analyses.  The core contribution is demonstrating that modelling a user's past opinions is a critical signal, on par with or even more informative than standard factors like demographics and ideologies. This insight could reframe debates in social sciences on whether ideologies alone capture individual variance. The authors thoroughly analyze different prompting combinations on multiple LLMs using the OpinionQA dataset. The results clearly show gains from adding past opinions to user profiles, with the full model reaching up to 7% higher accuracy. Additional BERT experiments confirm the consistency of these trends.

I think the paper is technically sound, with reproducible experiments and sufficient analysis, especially since the framing and motivation situate the work well within active discussions on model personalization in the NLP community.

In conclusion, the work provides valuable insights into modelling different user factors, laying the groundwork for personalized applications.